# Tart Cherry Extract and Omega Fatty Acids Reduce Behavioral Deficits, Gliosis, and Amyloid-Beta Deposition in the 5xFAD Mouse Model of Alzheimer’s Disease

**DOI:** 10.3390/brainsci11111423

**Published:** 2021-10-27

**Authors:** Zackary Bowers, Panchanan Maiti, Ali Bourcier, Jarod Morse, Kenneth Jenrow, Julien Rossignol, Gary L. Dunbar

**Affiliations:** 1Field Neurosciences Institute Laboratory for Restorative Neurology, Central Michigan University, Mt. Pleasant, MI 58859, USA; bower1zl@cmich.edu (Z.B.); maiti1p@ascenstion.org (P.M.); jenro1k@cmich.edu (K.J.); rossi1j@cmich.edu (J.R.); 2Program in Neuroscience, Central Michigan University, Mt. Pleasant, MI 48859, USA; 3College of Health and Human Services, Saginaw Valley State University, University Center, Saginaw, MI 48710, USA; alibourcier@gmail.com (A.B.); morsej@bgsu.edu (J.M.); 4Department of Psychology, Central Michigan University, Mt. Pleasant, MI 48859, USA; 5Field Neuroscience Institute Laboratory of Restorative Neurology, Ascension St. Mary’s Hospital, Saginaw, MI 48604, USA; 6College of Medicine, Central Michigan University, Mt. Pleasant, MI 48859, USA

**Keywords:** Alzheimer’s disease, neurodegeneration, inflammation, 5xFAD, total body rhythm, antioxidants, omega fatty acids

## Abstract

Combined treatments using polyphenols and omega fatty acids provide several therapeutic benefits for a variety of age-related disorders, including Alzheimer’s disease (AD). Previously, we found a commercial product, Total Body Rhythm (TBR), consisting of tart cherry extract, a potent polyphenol, and omega fatty acids, significantly reduced memory, and neuropathological deficits in the 192 IgG-saporin mouse model of AD. The present study assessed the efficacy of TBR for treating behavioral and neuropathological deficits in the 5xFAD model of AD. Both 6- and 12-month-old 5xFAD mice and age-matched wild-type controls received TBR (60 mg/kg) or the equivalent dose of vehicle (0.5% methylcellulose) via oral administration, every other day for two months. All mice were tested in the open field (OF), novel object recognition (NOR), and the Morris water maze (MWM) tasks. In addition, neuronal morphology, neurodegeneration, Aβ plaque load, and glial activation were assessed. TBR treatment reduced memory deficits in the MWM and NOR tests and lessened anxiety levels in the OF task, mostly in the 6-month-old male mice. TBR also protected against neuron loss, reduced activation of astrocytes and microglia, primarily in 6-month-old mice, and attenuated Aβ deposition. These results suggest that the combination of tart cherry extract and omega fatty acids in TBR can reduce AD-like deficits in 5xFAD mice.

## 1. Introduction

Alzheimer’s disease (AD) is a progressive neurodegenerative disorder and is the leading cause of senile dementia [1,2]. Deposition of amyloid plaques and hyperphosphorylated tau are the principal pathologies observed in the AD brain, along with increased oxidative stress and gliosis [3,4]. These factors are associated with synaptic loss, neuronal degeneration, and cognitive impairment in AD [5]. Several researchers have attempted to reduce these pathologies using small molecules, synthetic drugs, natural polyphenols, and immunotherapies; however, most of them show less than satisfactory outcomes. For example, some drugs were able to reduce amyloid beta protein (Aβ) plaques but were only able to marginally improve cognitive performance [6,7].

Several reasons could explain the failure to achieve the effective therapeutic effects of these drugs. Biochemical and cellular studies have demonstrated that accumulation of misfolded Aβ and hyperphosphorylated tau tangles in the AD brain can be exacerbated by or even result from increased oxidative stress [8]. Studies have shown these AD symptoms to be linked to deficits in essential nutrients such as vitamins with antioxidant properties as well as inadequate or unbalanced intake of omega fatty acids [9]. These previous research reports prompted us to investigate a proprietary combinatorial therapy of antioxidants and omega fatty acids in the 5xFAD model of AD.

Transgenic mouse models are crucial for understanding AD phenotypes. The 5xFAD TG6799 model used in this study has been characterized extensively [10,11]. Briefly, these mice have a total of five AD gene-linked mutations in the amyloid precursor protein (APP) and presenilin 1 (PSEN1). These mutations result in half the endogenous levels of APP and PSEN1 protein levels observed in human whole brain [11]. Intracellular amyloid beta accumulations start at around 1.5 months, with plaque formation and gliosis emerging at around 2 months of age [10]. Gliosis in the 5xFAD recapitulates the changes seen in humans, with amyloid accumulation and glial cell proliferation increasing with age and plateauing at around 10 months [12]. Female 5xFAD mice are known to exhibit more aggressive plaque pathology, which continues to increase until 14 months of age [12].

Gliosis was one of the three initial abnormalities identified by Alois Alzheimer in his original paper characterizing the pathology of AD [3]. In fact, subsequently, Alzheimer went on to edit an entire text on identifying heterogenous glial cell responses in pathological nervous tissue [13], and we are still seeking to unravel the fundamental roles of glial cells in AD [13,14]. Understanding the relationship between glial cell functioning, fatty acid intake, and exogenous antioxidants in both the healthy and AD brain is critical, as recent evidence indicates that astrocytes could take on a toxic gain of function role in AD [15].

DHA, an omega-3 fatty acid, is the primary fatty acid in the nervous system. Decreases in total polyunsaturated fatty acids in the AD brain have been linked to reduced NMDA receptor function and demyelination, possibly decreasing overall speed of neurotransmission and cognitive abilities [9,16]. Recently, Framingham and colleagues reported that there is a strong correlation between plasma DHA levels and dementia [17]. For example, patients with high plasma DHA levels had a substantially lower chance (47%) of developing dementia [17,18]. In addition, omega fatty acids showed positive effects in clinical trials of AD and cardiovascular diseases [2,18].

In 2006, the Kame project found that consuming fruit and vegetable juices three times per week may play an important role in delaying AD [19]. Nutrition-related risk factors, such as hyperglycemia, diabetes, hypercholesteremia and prehypertension, can begin decades before cognitive decline is apparent. Therefore, antioxidant therapy could be viable option for AD therapy. The main hurdles when using supplements is that the methods of processing them can affect the total levels of antioxidants available, as well as their bioavailability. It has been demonstrated that tart cherries maintain much of their antioxidant capabilities upon processing and that anthocyanin components are readily absorbed from the stomach of rodents [20,21].

In our previous research, we demonstrated that with administration of the Total Body Rhythm (TBR) product, a significant amount of several antioxidants, including anthocyanins, improved cholinergic neurons and protected against memory loss in novel object recognition (NOR) tasks in a 192 IgG-saporin-induced model of AD [22]. Our findings prompted us to investigate whether TBR could also reduce cognitive deficits and combat oxidative stress, neuroinflammation, and amyloid plaque load in the transgenic 5xFAD mouse model of AD.

## 2. Materials and Methods

### 2.1. Animals

All mice were kept on a reverse day/night cycle with 12:12 h. Two age groups (6 and 12 months) of 5xFAD mice were used for the current study. The mice at both ages were divided into four groups: wild-type mice receiving vehicle (WT+VEH), wild-type mice receiving TBR (WT+TBR), AD mice receiving vehicle (AD+VEH), and AD mice receiving TBR (AD+TBR). Gender differences were observed only for the NOR and immunofluorescence imaging of Aβ in this study, so for all other measures, the analyses for male and female were combined. The number of mice used for each procedure is shown in Table 1.

### 2.2. Genotyping

The 5xFAD and wild-type progeny were born to designated core breeder animals. Genotyping was conducted via tail snips when the mice were approximately three weeks of age (before weaning).

### 2.3. Treatments

TBR was generously donated by InTerra Nutraceuticals (Manistee, MI, USA). TBR is composed of tart cherry extract, containing Montmorency cherries with an anthocyanin content of 426.7 μg/mg and omega-3 fatty acids from mercury-free Nordic fish oil and omega-6 and omega-9 from (ultra-refined emu/kalayla oil). Both the 6-month-old and 12-month-old 5xFAD mice (*n* = ~10/group) were treated with TBR (60 mg/kg) suspended in 0.5% methyl-cellulose PBS solution. Administration was conducted by oral gavage, 24 h after baseline behavior and every other day for 2 months. Age-matched control mice received the equivalent amount of vehicle (0.5% methylcellulose in PBS). Dose was determined by the previously published work from our laboratory [22]. Timing of the treatment and testing procedure is depicted in Figure 1.

### 2.4. Behavioral Testing

#### 2.4.1. Open-Field Testing

Open-field (OF) tests were conducted one day prior to treatment and on day 64, twenty-four hours after receiving the last treatment. The OF testing was used to assess spontaneous motor activity and anxiety [23,24]. Additionally, fecal boli counts were taken during each trial. Mice were tested in one 30 min trial at both pre- and post-treatment. For details, see [24].

#### 2.4.2. Novel Object Recognition

The novel object recognition (NOR) task allows for the assessment of recognition memory in an environment with minimal stress [22,25,26,27]. We have published details of NOR procedure [24]. Mice were first acclimated for 5 min to the NOR apparatus, a gray polyvinyl box (40 cm × 40 cm × 40 cm). During training, the mice were given 10 min to explore the NOR containing two identical objects, designated the familiar objects (FO). Following a 5 min inter-trial interval in their home bins, the mice were placed back in the NOR for the acquisition trial. During the acquisition trial, one of the FOs was replaced with a novel object (NO) of similar size and interest (Figure 2).

#### 2.4.3. Morris Water Maze (MWM)

The MWM task measures the ability to find a hidden platform in a pool of water and is used as an assessment of procedural learning and spatial memory [24,28,29]. The MWM consists of a large circular water tank (180 cm in diameter) in which mice swim in 28 °C water, made opaque by the addition of non-toxic tempura paint. The mice attempt to locate a platform submerged 1 cm below the surface of the water. For analyses, the image of water tank was divided into 4 quadrants (NE, SE, SW, NW) with a digital camera supporting the ANY-Maze software in recording the movement of the mice. All mice were given one day of habituation, in which they were given two 1 min trials to acclimate to the pool and testing room. The day after habituation, the MWM testing was initiated, with each mouse being placed in the tank, facing the wall, in one of four randomly assigned starting positions (N, S, E, W). Each trial lasted until the mouse found the platform, or when a 60 s time limit was reached. If the mouse did not find the platform within 60 s, it was gently guided, by hand, to the platform and allowed to rest on it for 30 s. After a 10 min inter-trial interval, each mouse was re-tested. All mice were given 4 trials daily for 5 consecutive days. Following testing, the mice were towel-dried and placed in a warming bin before being returned to their home bin.

#### 2.4.4. MWM Probe Trial

Twenty-four hours following the fifth and final training session, the mice were tested for memory consolidation using a probe trial, in which the platform was removed from the pool and the swim patterns of the mice were recorded for the 60 s trial to determine the latency to enter the location where the platform was previously located [29].

#### 2.4.5. Reverse Trial

Twenty-four hours following the probe trial, the mice were tested on a reverse spatial task. During the reverse trial, the platform was relocated from the original site, i.e., SW to the NE quadrant, and mice were given 60 s to locate the platform. Four trials using all four starting points (within the order randomly determined) were conducted for the reverse learning, with a 10 min inter-trial interval. Latency (s) to locate the hidden platform was the primary dependent variable.

### 2.5. Euthanasia

Twenty-four hours after the last behavioral test, the mice were euthanized using either cervical dislocation for those brains used for Western blot or by overdose of sodium pentobarbital (Fatal Plus, 1 mL/4.53 kg) for those brains used for immunohistochemistry (IHC). Mice were perfused transcardially using phosphate-buffered solution (PBS), followed by 4% paraformaldehyde solution. After the brains were extracted, they were transferred to 4% paraformaldehyde for 24 h and then transferred into 1xPBS and stored at 4 °C for later use.

### 2.6. Tissue Analysis

#### 2.6.1. Cresyl-Violet Staining for Morphology

The brains from all groups were sectioned in the coronal plane at 40 μm, using a cryostat, before being stained with 0.1% cresyl-violet. The sections were washed, dehydrated with graded alcohol, cleared, mounted, and cover slipped. Photomicrographs were taken using a compound light microscope (Olympus, Shibuya-ku, Tokyo, Japan) with a 40× objective (total magnification of 400×). The number of pyknotic, or tangle-like, cells were counted manually and were expressed as number of pyknotic cells per microscopic field. A minimum of 3 different sections from each brain area, per mouse, were used for counting the number of pyknotic cells in each group.

#### 2.6.2. Amyloid β-Plaque Count

For both 6- and 12-month-old mice, coronal sections were sampled from bregma at –1.28 mm and −2.92 mm. Sections were stained using curcumin, which labels plaques as efficiently as Aβ-specific antibodies [30]. Using a 24-well plate, 40 μm sections were washed for 5 min with double-distilled water and dehydrated with graded alcohol (at 50% and 70%) for 2 min each. Then, the sections were labeled with curcumin (10 μM) dissolved in 70% ethanol (ETOH) for 1 min, followed by a 1 min wash in 70% ethanol (ETOH). Tissue sections were then mounted using aqueous anti-fading media and cover slipped. All fluorescent imaging was performed using an Olympus BH-2 microscope using 40× objectives (total magnification = 400×). Values are expressed as the average number of Aβ plaques per microscopic field. A minimum of 5 serial sections, with 7–10 randomly selected different fields in 3 locations were used to quantify Aβ plaques. The mean from each group (*n* = ~5/group) was calculated from the counts of two researchers who were blind to treatment group identities.

#### 2.6.3. Western Blot

For Western blot analyses, a small portion (20–50 mg) of cortex and hippocampal frozen brain tissue was homogenized with a motorized tissue homogenizer in radio-immunoprecipitation assay (RIPA) buffer (10 mM Tris-Cl (pH 8.0), 1 mM EDTA, 0.5 mM EGTA, 1% Triton X-100, 0.1% sodium deoxycholate, 0.1% SDS, 140 mM NaCl, (pH 7.4) with protease and phosphatase inhibitors (Sigma, St. Louis, MO, USA). Following homogenization, the cell lysate was centrifuged at 13,000× *g* for 30 min at 4 °C. Following centrifugation, the supernatant was collected and aliquoted in small PCR tubes and then stored at −80 °C, until use. Total protein concentrations for individual samples were determined using the Pierce BCA protein assay kit (Thermo Scientific, Rockford, IL, USA). Samples were added with an equal amount of 2 × SDS sample buffer (125 mM Tris-HCl, pH 6.8, 4% sodium-dodecyl-sulfate, 20% glycerol, and 10% 2-mercaptoethanol) and then heated at 100 °C for 2 min and allowed to cool at room temperature. Approximately 150 µg of protein, per lane, was loaded and electrophoresed on 4–20% Tris-glycine gel, and then transferred to PVDF membrane (Millipore, Bedford, MA, USA). After incubating with mouse GFAP Antibody (GA-5): sc-58766 Santa Cruz Biotechnology (1:1000) or Iba1 Invitrogen (PA5-21274) (1:2000) for overnight at 4 °C on the shaker, the blots were washed three times in TBST and probed with anti-rabbit (sc-2357 Santa Cruz Biotechnology) or anti-mouse (sc-358917, Santa Cruz Biotechnology) secondary antibodies (1:2000) conjugated with horseradish peroxidase (HRP) for 1 h at room temperature. Then the blots were developed with ImmobilonTM Western Chemiluminescent HRP-substrate (Millipore, Billeria, MA, USA). The relative optical density was measured using ImageJ software (https://imagej.nih.gov/ij/ accessed 28 December 2020). To ensure equal protein loading in each lane, the blots were re-probed with GAPDH antibody ((D16H11) Cell Signaling Technology (1:1000)).

### 2.7. Data Analyses

Experimenters were blinded to treatment conditions during testing and data acquisition. All data are expressed as mean ± SEM. Weight changes and training for the MWM were analyzed using a two-way repeated measures analysis of variance (ANOVA). Between-group differences for OF, NOR, and MWM testing (including the probe and reversal tests) were all analyzed using one-way analysis of variance (ANOVA), followed by Tukey HSD (Honestly Significant Difference) test post hoc analysis, when appropriate. Probability of ≤0.05 was considered as statistically significant.

## 3. Results

### 3.1. Weight

Repeated measures ANOVA revealed no significant changes in weight of both the 6-and 12-month-old mice before and after treatment (Figure 3).

### 3.2. Open Field Test

A one-way ANOVA revealed no significant differences between 5xFAD and WT mice on total distance travelled (cm) or their average speed (cm/s) in the open field (OF). There was a significant difference for the 6-month-old mice for the average number of fecal boli per group (*F*(3, 33) = 4.71, *p* < 0.05), (Figure 4). Using a Tukey HSD post hoc analysis, we found that there was a significant difference in genotype effect for the 6-month-old mice (*p* < 0.05). There was also a significant treatment effect between the 5xFAD mice treated with vehicle and the 5xFAD mice treated with TBR (*p* < 0.05). There were no significant differences between the 12-month-old mice on counts of fecal boli.

### 3.3. Novel Object Recognition

A one-way analysis of variance (ANOVA) revealed a significant difference between the 6-month-old groups (*F*(3, 35) = 3.6377, *p* < 0.05) on the total time spent exploring the novel object (Figure 5). A Tukey’s HSD post hoc analysis revealed significant differences between the AD+VEH and AD+TBR mice (*p* < 0.05). There were no significant between-group differences in the 12-month-old mice.

After analysis of sex differences, we found that only the male 6-month-old mice spent significantly more time exploring the novel object (*F*(3, 18) = 3.3188, *p* < 0.05) than female mice (Figure 6).

### 3.4. Morris Water Maze

A repeated measures ANOVA for 6-month-old mice revealed a significant within-group effect for latency (s) to locate the escape platform over testing days, (*F*(4, 144) = 13.671, *p* < 0.001), but no significant between-subjects differences were found for treatment or interaction of treatment with time (Figure 7A). For the 12-month-old animals, there was a significant main effect for within-group difference across training days (*F*(4, 104) = 4.11, *p* < 0.05). There was a significant between-subjects effect (*F*(3, 27) = 5.593, *p* < 0.001) but not a significant interaction between days and treatment (Figure 7B). There was a significant difference in interaction between days and treatment for the 12-month-old WT+VEH mice and the 5xFAD+VEH (*p* < 0.05) and 5xFAD+TBR (*p* < 0.05) mice for day 3. There was a significant difference between the 12-month-old WT+VEH and 5xFAD+VEH mice for days 4 and 5 (Figure 7B). There was not a significant difference between the WT+VEH mice and WT+TBR and 5xFAD+TBR mice for days 4 and 5. These data indicate that the learning curves were similar for all mice in the 6-month-old group, although mice in the 12-month-old groups WT+VEH, WT+TBR, and 5xFAD+TBR learned the task faster than the 5xFAD+VEH mice.

A one-way ANOVA revealed no significant difference for latency (s) to the escape platform for either 6- or 12-month-old mice on the probe trial (*F*(3, 29) = 1.4152, *p* > 0.05 and *F*(3, 42) = 0.6166, *p* > 0.05, respectively).

Using a one-way ANOVA, we found that there was a significant difference in latency to locate the escape platform on trial one of reversal training for 6- and 12-month-old mice (*F*(3, 26) = 3.134, *p* < 0.05 and *F*(3, 21) = 5.486, *p* < 0.0001, respectively) (Figure 8). Tukey’s HSD analysis revealed a significant difference between the 6-month-old 5xFAD+VEH and WT+TBR animals (*p* = 0.013) and between 12-month-old 5xFAD+VEH vs 5xFAD+TBR and WT+TBR mice (*p* = 0.0078 and *p* = 0.0183).

### 3.5. Cresyl-Violet Imaging for Neuronal Morphology

Neuron morphology was assessed by staining sections with 0.1% cresyl-violet and counting the number of pyknotic neurons in the CA1 and CA3 areas of the hippocampus, as well as in the pyramidal layer 5 of the PFC (Figure 9A). Using a one-way ANOVA, it was revealed that there was a significant decrease in the number of pyknotic cells in 6-month-old 5xFAD mice treated with TBR in area CA1 (*F*(3, 46) = 6.6899, *p* < 0.05) (Figure 9B), CA3, and PFC (*F*(3, 25) = 12.8155, *p* < 0.05) (Figure 9D).

### 3.6. Amyloid β-Plaque Staining

A two-way independent *t*-test revealed a significant difference in the average number of Aβ plaque for the 6-month-old mice in the retro-splenial cortex between the four AD 6-month-old mice treated with TBR and the five AD mice treated with vehicle (Figure 10B). There was a significant difference in the average number of β-amyloid peptide between the five AD 12-month-old mice treated with TBR and the six AD mice treated with vehicle (Figure 10C). There were no significant differences in the dentate gyrus for either age group in the number of Aβ plaques. However, subsequent analyses indicated that this difference was due primarily to how the male mice responded to TBR treatment (Figure 11).

### 3.7. Sex Differences in Number of Amyloid-β Plaques in 6-Month-Old Mice

One-way ANOVAs revealed significant sex differences in the average number of Aβ plaques for the 6-month-old 5xFAD mice. There was a significant difference in the number of Aβ plaques retrosplenial cortex (RSC) (*F*(3, 68) = 17.7579, *p* < 0.05), dentate gyrus (DG) (*F*(3, 72) = 10.4856, *p* < 0.05), and entorhinal area (EC) (*F*(3, 67) = 8.4006, *p* < 0.05) (Figure 11A–C). Using the Scheffè post hoc analysis, it was found that there was a significant difference in the number of Aβ plaques between the male 5xFAD mice treated with TBR and the male 5xFAD+VEH, female 5xFAD+VEH, and the female 5xFAD+TBR (*p* < 0.05). The 6-month-old female 5xFAD mice treated with TBR were not significantly different than the female mice treated with vehicle (Figure 11A,B). Using the Scheffè post hoc analysis, it was found that there was a significant difference between the male 5xFAD mice treated with TBR, female 5xFAD+VEH, and female 5xFAD+TBR (*p* < 0.05) in the DG. The 6-month-old female 5xFAD mice treated with TBR were not significantly different than the female mice treated with vehicle (Figure 11C).

### 3.8. Western Blot

One-way ANOVA revealed significant differences in optical density between 6-month-old animals for GFAP in the hippocampus and cortex in comparison with WT mice (*F*(3, 25) = 45.3889, *p* < 0.001) (Figure 12A–C). There was a significant difference in cortex levels of GFAP for 6-month-old mice only (*F*(3, 12) = 22.53, *p* < 0.001). Tukey’s HSD test revealed a significant difference between the 6-month-old 5xFAD+VEH group and all other groups (*p* < 0.05) in the hippocampus. There was not a significant difference between the 5xFAD+TBR mice and the WT+VEH or WT+TBR mice in the hippocampus or cortex.

One-way ANOVAs revealed significant differences in optical density in hippocampal (*F*(3, 16) = 28.4157, *p* < 0.001) and cortical (*F*(3, 14) = 8.95, *p* < 0.05) Iba1 levels in the 6-month-old mice. Tukey’s HSD test revealed a significant difference between the 6-month-old AD+VEH group and all other groups (*p* < 0.001) in the hippocampus and cortex. There was no significant difference between the AD+TBR mice and the WT+VEH or WT+TBR mice, indicating an intermediate effect. There was not a significant difference between the 12-month-old mice in the hippocampus or cortex.

## 4. Discussion

The current study tested the combinatorial treatment of tart cherry extract and omega fatty acids (TBR) in 5xFAD AD mice at both 6 and 12 months of age. Our findings confirmed our hypotheses that mice receiving TBR would have fewer cognitive deficits, increased neuron protection, changes in glial cell functioning, and reduced β-amyloid load. Previous work from our lab has demonstrated that TBR can prevent loss of body weight in a cholinergic-toxin (192 IgG-saporin) mouse model of AD [22]. The finding that there were no significant differences between the 5xFAD animals and age-matched WT controls in weight or of activity in our study underscores the argument that the differences seen in the NOR and MWM tasks were due to disturbances in cognition rather than overall decrements in health or activity levels [31]. However, the findings of an increased level of anxiety in untreated 5xFAD mice may have influenced the outcome of the NOR tasks, although differences in the number of approaches to the objects were not observed, suggesting that the differences in the amount of time investigating the novel object were likely due to recognition memory differences rather than differences in anxiety levels or motivation.

We were able to demonstrate that 6-month-old 5xFAD mice that were treated with TBR had preservation of recognition memory by spending increased time with the novel object during NOR testing (Figure 5). The 12-month-old 5xFAD mice treated with TBR were not significantly different than the 5xFAD mice treated with vehicle, although there was a similar data distribution to the 6-month-old mice on total time spent with the novel object. In addition to sparing recognition memory, as measured in the NOR, the 5xFAD mice treated with TBR had improved performance on a test of spatial memory using the MWM (Figure 8). During training on the MWM, all mice learned to perform the task; however, there was a significant difference between the 6- and 12-month-old AD mice in latency to locate the platform across training days, with 12-month-old AD mice taking more time to find the hidden platform. Nonetheless, during the probe trial, there were no significant between-group differences in latency to enter the quadrant where the platform was previously located, suggesting that all mice had learned the platform location by the end of training. However, during a reverse trial in which the platform was relocated to the opposite quadrant from its original location, there were significant deficits in both 6- and 12-month-old vehicle-treated 5xFAD mice but not in mice treated with TBR.

Multiple mechanisms could explain why the 12-month-old 5xFAD mice treated with TBR had increased ability to locate the escape platform compared with vehicle-treated mice. For example, Siwek and colleagues have demonstrated that the genes that are persistently altered in the 5xFAD mouse model are genes related to inflammation and that these changes can persist past one year of age [32]. It has been demonstrated repeatedly that diets high in antioxidant foods and, specifically, anthocyanins, have been linked to slowing age-related cognitive decline [33,34]. Additionally, AD transgenic animals are known to have decreased ability to locate the escape platform during the MWM, and these deficits are exacerbated by depletion of DHA, and it has been demonstrated that supplementation ameliorated the deficits seen in AD animals on the MWM [16].

In addition to cognitive sparing, we found that TBR was able to reduce the number of pyknotic cells in the hippocampus and prefrontal cortex of the mouse brain (Figure 9). Alzheimer’s disease, unlike normal aging, is associated with a large loss of neurons [35]. Furthermore, the total number of lost neurons is positively correlated with severity of symptoms [36]. Pyknotic cells are those which are undergoing necrosis or apoptosis. For both 6- and 12-month-old mice, there were reduced numbers of pyknotic cells in those animals that received TBR treatment. It is possible that TBR, through reducing oxidative stress and correction of fatty acid profiles, was able to protect both neurons and glial cells.

We demonstrated that glial cell activity was reduced in the 6-month-old 5xFAD mice treated with TBR, including GFAP for astrocyte proliferation and IBA1 for gross microglial activity in both the hippocampus and cortex. It is known that both microglia and astrocytes are intimately involved with β-amyloid [15,37]. This finding has been supported by multiple studies in recent decades [15,38]. Researchers demonstrated that not only were glial cells found in proximity to β-amyloid and tau but that microglia and astrocytes can have both a formative or detrimental effect and a neurodegenerative or preventative effect on the symptoms of AD. Resveratrol, the popular antioxidant in red wine, is believed to possibly exert neuroprotective effects through inhibition of microglia-dependent α-amyloid toxicity in AD models [39]. We have demonstrated that TBR reduced both GFAP and microglial activity in the 6-month-old mice treated with TBR but that it did not have the same effect in the 12-month-old mice. It is known in both the human AD brain and in the 5xFAD mouse model that glial cells are increased [10]. We do not know if gliosis is detrimental, as astrocytes could increase synapse formation to maintain ion homeostasis [14] or could increase the amount of growth factor supplied to neurons. Our findings suggest that early intervention with TBR may be necessary to slow the progression of astrocytic and microglial activation that is often associated with disease-induced brain states, but further research is needed to elucidate the extent to which gliosis is detrimental vs. beneficial for restoring neuronal functioning.

The 5xFAD mouse model is known to be an aggressive model, with rapid formation of amyloid plaques. We demonstrated that TBR was able to reduce amyloid beta levels within the retrosplenial cortex of the 6-month-old AD mice treated with TBR and the entorhinal cortex of 12-month-old mice. The retrosplenial cortex (RSC) constitutes a large portion of cortex in rodents, which corresponds to Broadman’s area 29 and 30 [40]. Although the RSC was identified by Broadman over 90 years ago, the structure and function of this region remain elusive. It is known that in sporadic AD there are volumetric differences in the RSC that are comparable with the atrophy seen within the hippocampus of humans [41]. It has been demonstrated that there are dramatic differences in size between species, and there is now a strong connection between the RSC cortex and a range of cognitive functions. According to an extensive review conducted by Vann and colleagues, the RSC has emerged as a key member of a network of brain regions, including the hippocampus and limbic system, and is involved in episodic memory, navigation, imagination, and planning for the future [40]. All of these functions are crucial for performing the memory tasks assessed in our study and could explain why we found such a significant difference at 48 h post-training in the MWM.

When exploring the impact that sex has on AD symptoms, we found a difference between the 6-month-old male and female 5xFAD mice on the NOR task and the number of amyloid β deposits. We found that male mice responded better to treatment on the NOR (Figure 6) than the female mice, in that they spent increased time exploring the novel object during testing. We also demonstrated that 6-month-old male mice had fewer amyloid β deposits. Understanding the relationship between amyloid β and cognitive impairment in AD is complex—some argue that there is very little connection, indeed patients can have varying degrees of cognitive deficits, even when having similar amyloid beta loads [42].

Our results show that TBR influences amyloid beta deposition in a sex-dependent fashion. It has been demonstrated in multiple transgenic mice, including APPswe, PS1 double transgenic, and 3xTg mice, that female mice have increased amyloid β burden compared with male mice [43,44]. This increased burden of amyloid-β was also found to coincide with worse cognitive outcomes for female mice as well [44]. In addition, it has been demonstrated for over two decades that both macaque and human males respond preferentially to donepezil, one of the most common cholinesterase inhibitors [45,46]. Both polyphenols and omega fatty acids, through different mechanisms, have been shown to reduce amyloid β load. However, the degree to which amyloid β load is related to cognitive deficits is debatable, but our findings that males had reduced levels of amyloid β and responded more strongly to TBR therapy suggests that increased amyloid β may have contributed to the memory deficits observed in this study.

Manipulation of sex hormone levels in the mice could be contributing to the sex differences observed in the behavioral and amyloid-β data of our study. It has been demonstrated that 5xFAD female mice can have increasing amyloid burden until 14 months of age, possibly due to influences of the Thy-1 promoter used to express the transgenes in this model of AD [11]. We hypothesize that either the tart cherry extract or the fatty acids influenced sex hormone differences occurring earlier in disease development, which then influenced outcomes; it is well known that within the human population and in animal models that females have higher rates of AD and that this difference cannot be attributed to simply living longer [47]. A recent comprehensive review using over 275 articles and meta-analysis found that women are more likely to suffer increased cognitive deterioration than men who are at the same AD disease stage [48]. Indeed, the same group who found a disparity between men and women in AD have found that episodic memory is a key factor in this discrepancy [49]. However, if the difference seen in the NOR in our study was primarily because of amyloid-β load, it could be argued that 12-month-old 5xFAD mice would be expected to have more amyloid-β than a 6-month-old female.

## 5. Conclusions

In our study, the beneficial effects of TBR, which contains a combination of fatty acids and antioxidants, which were shown previously in the 192 IgG-saporin mouse model of AD, were extended to the 5xFAD transgenic AD model. Our findings suggest that TBR may help reduce AD-induced memory loss and neuropathology and could serve as a valuable adjunctive therapy for AD. Further research is needed to adequately discern the degree to which each TBR ingredient contributed to the overall effect, but collectively TBR proved to be effective in reducing memory neuropathological deficits in the 5xFAD transgenic mouse model of AD.

## Figures and Tables

**Figure 1 brainsci-11-01423-f001:**
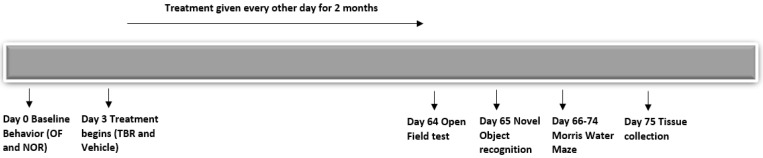
Experiment paradigm and treatment regimen. All mice were randomly divided into four groups (Table 1) and baseline behavior was recorded before and after the treatment of TBR (60 mg/kg BW), orally, every other day for two months.

**Figure 2 brainsci-11-01423-f002:**
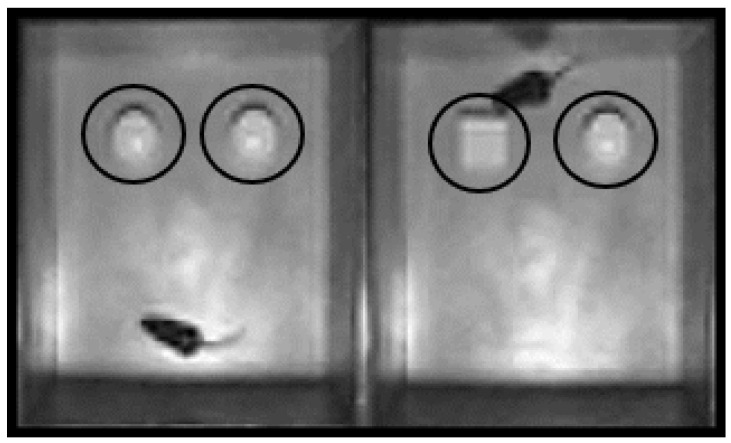
Novel object recognition. The placed objects were 7.5 cm from the side of the walls and 10.5 cm from each other. The region of interest was designated as the area surrounding the object and within 2 cm of the object. The ANY-Maze software measured the total number of nose pokes into the region of interest (ROI) and time spent exploring the objects.

**Figure 3 brainsci-11-01423-f003:**
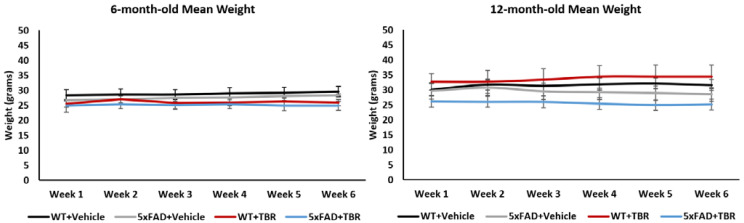
Mean weight. No between- or within-group differences in body weight were observed over the course of the study.

**Figure 4 brainsci-11-01423-f004:**
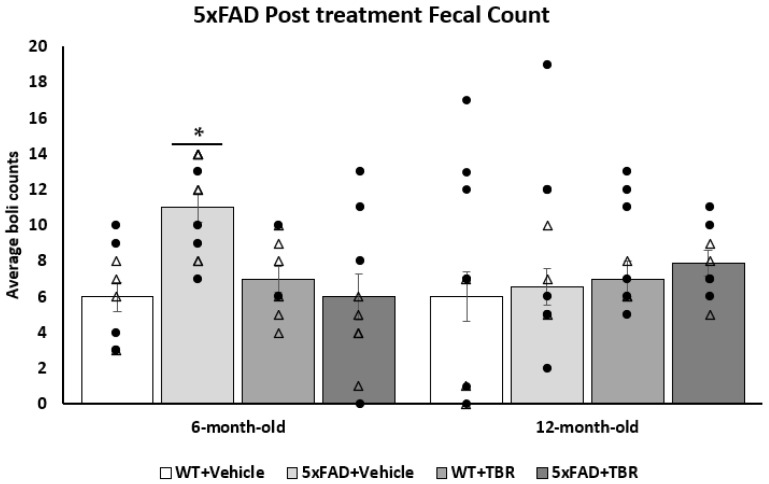
Effect of TBR on fecal boli in 5xFAD mice. Total number of fecal boli count in 6-month-old and 12-month-old mice treated with TBR. There was a significant increase in the number of fecal boli post-treatment excreted during open-field testing for the 6-month-old 5xFAD+VEH mice compared with the WT+VEH and 5xFAD+TBR mice. There were no significant differences in the number of fecal boli in 12-month-old mice. There was not a significant difference between the 5xFAD+TBR mice and the WT+VEH or WT+TBR mice. * *p* < 0.05 compared with WT+VEH, WT+TBR and 5xFAD+TBR. (∆, female mice; ●, male mice).

**Figure 5 brainsci-11-01423-f005:**
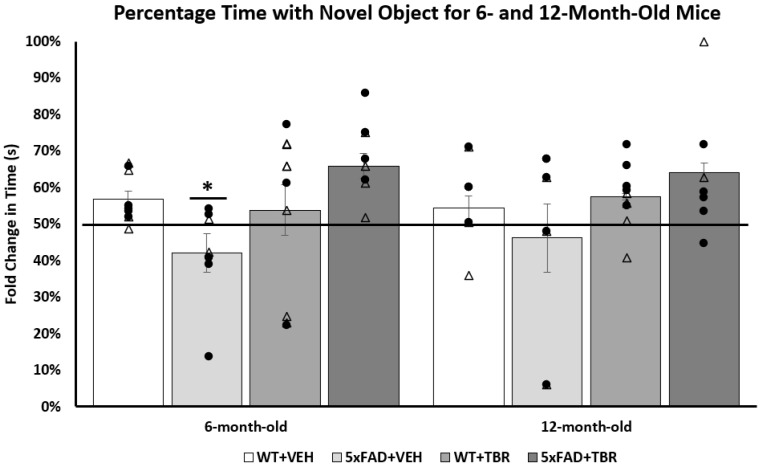
TBR-treated 6-month-old 5xFAD mice had reduced memory deficits in the novel object recognition task. The 6-month-old 5xFAD vehicle-treated mice spent a significantly shorter percentage of time exploring the novel object than all other groups. There were no significant between-group differ-ences in the 12-month-old mice. * *p* < 0.05, compared with all other groups. (∆, female mice; ●, male mice).

**Figure 6 brainsci-11-01423-f006:**
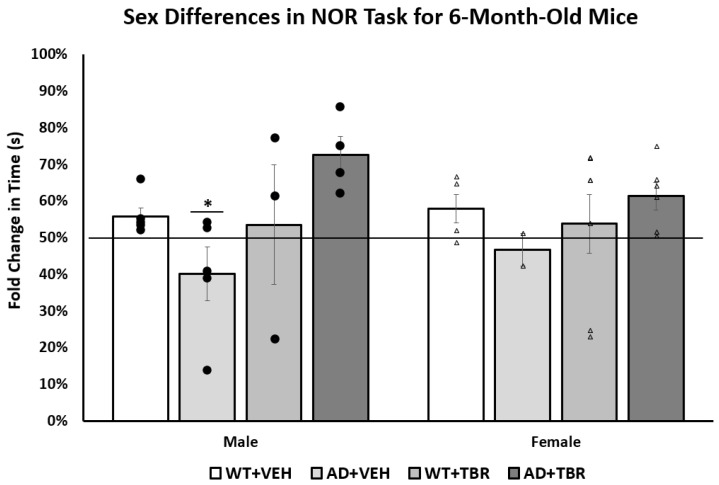
Male TBR-treated 6-month-old 5xFAD mice had fewer memory deficits in the novel object recog-nition task than female mice. There was a significant difference in the total time spent with the novel object for the 6-month-old male mice. Tukey’s HSD test revealed a significant difference between 5xFAD+VEH animals and 5xFAD+TBR (*p* = 0.05). There was no significant difference between the 6-month-old female mice. * *p* = 0.05 when compared with 5xFAD+VEH with 5xFAD+TBR mice. (∆, female mice; ●, male mice).

**Figure 7 brainsci-11-01423-f007:**
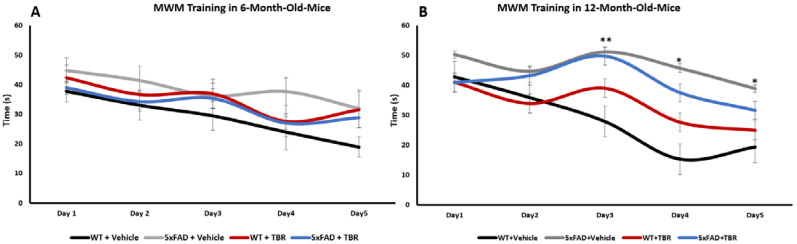
Effect of TBR on training of 5xFAD mice in the MWM task. (**A**) For the 6-month-old mice there was a significant within groups difference across training days, but not a significant difference between groups. (**B**) There was a significant genotype effect by day 3 of testing in the 12-month-old mice, with WT+VEH mice taking less time to find the hidden platform in the MWM task than either the 5xFAD+VEH (*p* < 0.05) or 5xFAD+TBR mice, but by days 4 and 5, only the 5xFAD+VEH mice took significantly longer to find the platform. ** *p* < 0.05, between WT+VEH mice and both groups of 5xFAD mice; * *p* < 0.05, between WT+VEH and 5xFAD+VEH mice.

**Figure 8 brainsci-11-01423-f008:**
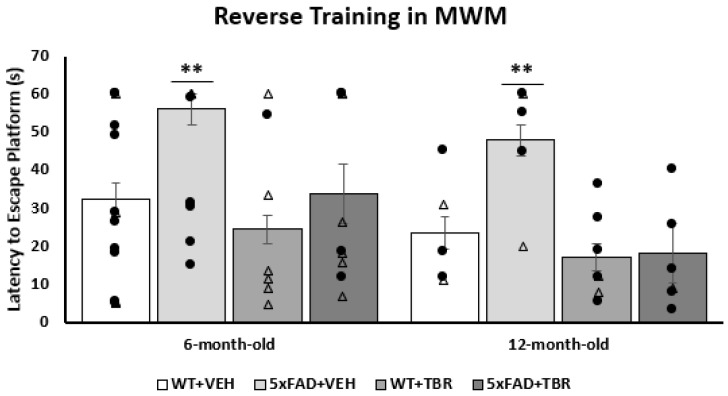
Effect of TBR on latency to find the platform during the first trial of reversal testing in 5xFAD mice. TBR treatments reduced spatial learning deficits for both 6- and 12-month-old 5xFAD mice during the platform reversal trial 1 on the MWM task. There was not a significant difference between the 5xFAD+TBR mice and the WT+VEH or WT+TBR mice. ** *p* < 0.01, compared with all other groups. (∆, female mice; ●, male mice).

**Figure 9 brainsci-11-01423-f009:**
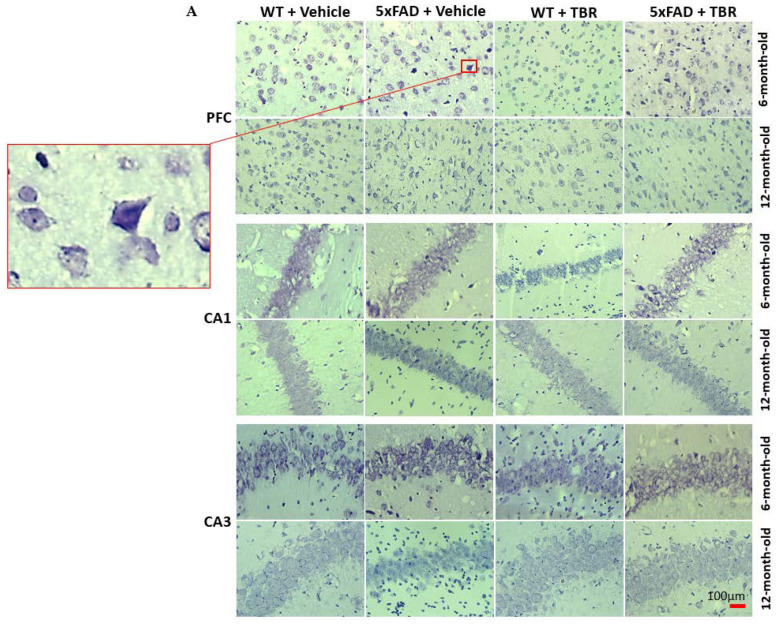
TBR treatment decreased pyknotic and tangle-like cells in the PFC and hippocampus of 5xFAD mice. (**A**) The 6- and 12-month-old 5xFAD and age-matched control mice were treated with (60 mg/kg BW) or vehicle for 2 months. (**B**) There was a significant difference in the number of pyknotic cells for 6- and 12-month-old mice in the PFC. (**C**) There was a significant difference in the number of pyknotic cells for 6- and 12-month-old mice in the CA1 area of the hippocampus. (**D**) In the CA3 area of the hippocampus there was a significant difference in the number of pyknotic cells for the 6-month-old animals but not for the 12-month-old mice. * *p* < 0.05.

**Figure 10 brainsci-11-01423-f010:**
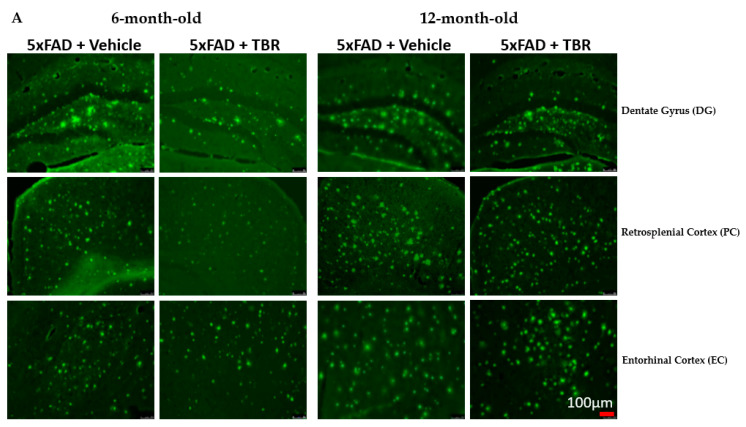
(**A**–**C**). Effect of TBR on Aβ plaque deposits in 5xFAD mice. (**A**) Representative image from three locations: the dentate gyrus (DG), retrosplenial cortex (RSC), and the entorhinal cortex (EC) for 6- and 12-month-old animals. (**B**) Using a two-tailed *t*-test, there was a significant difference in total amyloid β deposits in the retrosplenial cortex between 5xFAD 6-month-old mice treated with TBR and 5xFAD mice treated with vehicle. (**C**) Using a one-tailed *t*-test, there was a significant difference in total amyloid β deposits between 5xFAD 12-month-old mice treated with TBR and 5xFAD mice treated with vehicle. * *p* < 0.05.

**Figure 11 brainsci-11-01423-f011:**
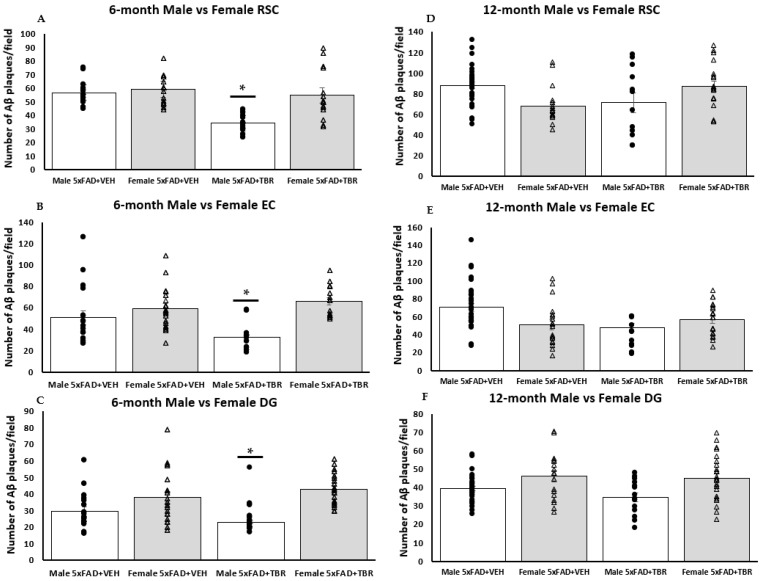
TBR reduced the number of amyloid β deposits in male 5xFAD mice. The 6-month-old male 5xFAD mice treated with TBR had significantly less amyloid β in the retrosplenial cortex (RSC) than did male 5xFAD mice treated with vehicle and female AD mice treated with TBR or vehicle (**A**). The 6-month-old male 5xFAD mice treated with TBR had significantly less amyloid β in the entorhinal cortex (EC) than male 5xFAD mice treated with vehicle and female 5xFAD mice treated with TBR or vehicle (**B**). The 6-month-old male 5xFAD mice treated with TBR had significantly less amyloid β than either group of 5xFAD female mice (**C**). There were no between-group sex differences in amyloid beta in the 12-month-old mice (**D**–**F**). * *p* < 0.05. (∆, female mice; ●, male mice).

**Figure 12 brainsci-11-01423-f012:**
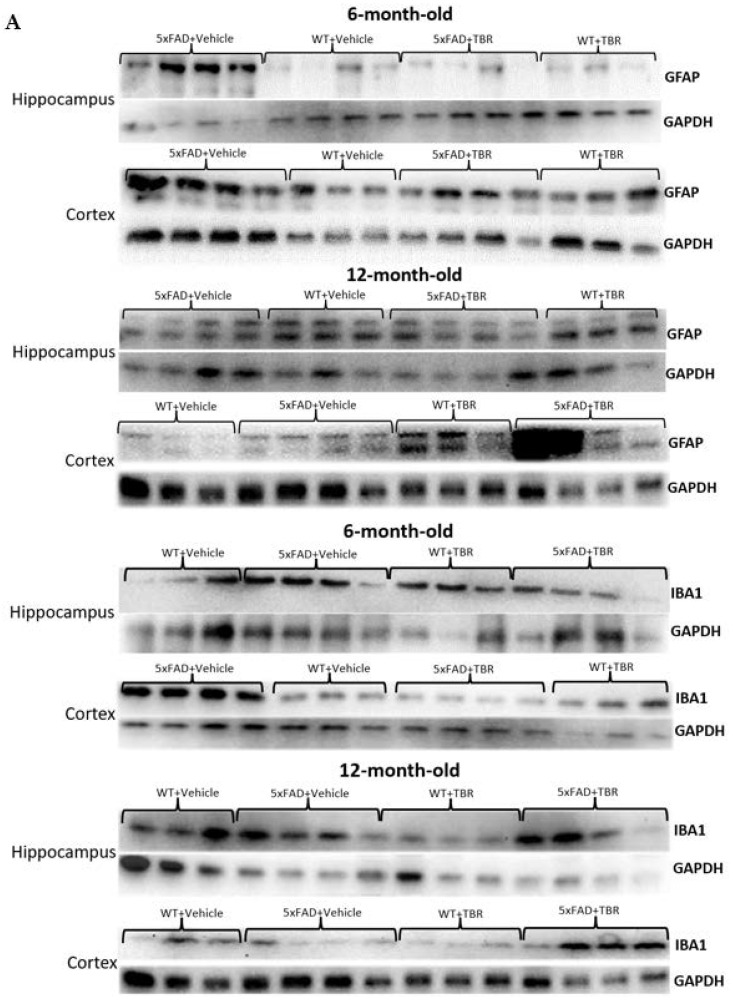
Western blots of GFAP and Iba1 levels in 6- and 12-month-old mice (**A**). There were between-group significant difference in hippocampal and cortical GFAP and Iba1 levels in the 6-month-old mice, but no significant between-group differences were observed in these measures for the 12-month-old mice (**B**–**E**). * *p* < 0.05; ** *p* < 0.05 for the 5xFAD mice, when compared with all other groups. Blots are representative images.

**Table 1 brainsci-11-01423-t001:** Experimental groups. Schematic diagram showing experimental design and treatment paradigm. Distribution of 6- and 12-month-old 5xFAD and age-matched WT mice for each procedural group.

Groups	Age	Gender	Behavior	Amyloid Beta	Western Blot	Cresyl-Violet
WT+VEH	6 months	M 6, F 6	12	6	5	3
AD+VEH	6 months	M 6, F 4	10	4	5	3
WT+TBR	6 months	M 3, F 7	10	4	5	3
AD+TBR	6 months	M 7, F 4	11	5	5	3
WT+VEH	12 months	M 6, F 4	10	5	3	3
AD+VEH	12 months	M 6, F 4	10	5	4	3
WT+TBR	12 months	M 6, F 4	9	5	3	3
AD+TBR	12 months	M 5, F 5	10	5	4	3

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
