# Peer review of "Tart Cherry Extract and Omega Fatty Acids Reduce Behavioral Deficits, Gliosis, and Amyloid-Beta Deposition in the 5xFAD Mouse Model of Alzheimer’s Disease"

_brainsci, 2021, doi:10.3390/brainsci11111423_

Round 1

Reviewer 1 Report

Bowers et al., convincingly demonstrate that treatment with Total Body Rhythm (which contains a combination of omega fatty acids and polyphenol from tart-cherry-extract) reduces several aspects of the pathology associated with the 5xFAD mouse model. The effects of treatment are stronger when treating from 4-6 months compared to 10-12 months and interesting sex differences are observed.

The experimental design seems solid and the manuscript is well written. Only minor corrections and considerations are need for this manuscript to be beneficial for the field.

Comments:

  • The authors should state whether the experimenter(s) were blinded to the treatment groups when conducting the experiments. 
  • According to Table 1; five mice from each group were used for western blot. However, fewer samples are depicted in Figure 11. The authors should clarify whether these blots are representative examples or correct the n values. The authors should also clarify what is displayed in the top blot (Fig 11A).  
  • The authors do a good job introducing the importance of omega fatty acids. But rather than focusing on the impact that it might have on microglial and astrocyte biology, they highlight potential beneficial impacts on protecting myelin and reducing demyelination. Given the decision to experimentally focus on microglia (Iba1) and astrocytes (GFAP). The authors should add more weight to gliosis in the introduction. Although, they do touch on it briefly in the discussion. Adding more data on how myelin levels are affected across the treatment groups would additionally strengthen the manuscript.
  • The claim that decreases in polyunsaturated fatty acids are linked to reduced demyelination (second paragraph of the introduction) is not supported by reference [7]. This needs to be corrected.
  • Several of the statements in the first paragraph of the introduction needs to be supported by citations.
  • The consistent color scheme applied through the figures, for the different treatment groups, works well. However, it would be informative for the reader and beneficial for transparency if the value for each individual mouse were represented on the bar graphs as well. Ideally, males and females could be displayed by different icons across all the bar graphs. Especially given the interesting sex differences that the authors report. 

Reviewer 2 Report

The authors examined the beneficial effects of tart cherry extract and omega fatty acids in the 5xFAD mouse model of alzheimer's disease. The authors primarily assessed the behaviors and neuronal structure in the 6-month-old and 12-month-old mice treated with or without the TBR. This is an interesting study particularly on the use of natural ingredients such as TBR as therapy in ALzheimer's disease.

However not all data for 12-month-old were presented although they mentioned that there is no significant difference. All data should be provided whether there is any significant difference or not.
Also, no discussion on the use of both 6-month- and 12-month-old mice, although it is clear that 12-month-old mice would theoretically provide evidence of Alzheimer's disease although this needs to be validated in the current study.

The authors should discuss the age-dependent effects seen in the mice and if (and how) TBR treatment improved the progression of the disease.

Overall, the manuscript is easy to understand and follow. The data corresponding to the body weight (for results section 1) should be provided as well. 

Round 2

Reviewer 2 Report

The authors have addressed most of my concerns. Some minor comments are:

1) Almost in all figures, please describe the symbols (closed black circle and open triangle) in the figure legend.

2) FIgure 5: y-axis label should be fold change in time (since the values plotted are less than 1). The authors mentioned "percent time" so the y-axis values should be between 0-100

3) FIgure 6: y-axis label should be fold change in time (since the values plotted are less than 1). The authors mentioned "percentage total time" so the y-axis values should be between 0-100

4) Figure 9: are you showing the number of cells or % of cells?

5) Line 400: "...for the 12-mont-old mice." should be "12-month-old"

6) Line 377: "....no between-groupt...." should be "....no between-group"

Author Response

Thank you for your careful edits and the helpful insights on this manuscript.  We have made all the corrections you suggested in this new revised manuscript.
